# Myeloid-Specific Pyruvate-Kinase-Type-M2-Deficient Mice Are Resistant to Acute Lung Injury

**DOI:** 10.3390/biomedicines10051193

**Published:** 2022-05-21

**Authors:** Xinlei Sun, Fujie Shi, Weiran Wang, Yunfei Wu, Shuang Qu, Jing Li, Hongwei Liang, Ke Zen

**Affiliations:** 1State Key Laboratory of Pharmaceutical Biotechnology, School of Life Sciences, Nanjing University, 163 Xianlin Avenue, Nanjing 210023, China; 1620204596@cpu.edu.cn (X.S.); 15951606225@163.com (F.S.); wangweiran0130@163.com (W.W.); jingli220@nju.edu.cn (J.L.); 2School of Life Science and Technology, China Pharmaceutical University, 639 Longmian Avenue, Nanjing 211198, China; 17302593369@163.com (Y.W.); qushuang171116@163.com (S.Q.); lianghongwei0418@163.com (H.L.)

**Keywords:** PKM2, neutrophil, degranulation, inflammation, acute lung injury

## Abstract

Infiltration of polymorphonuclear neutrophils (PMNs) plays a central role in acute lung injury (ALI). The mechanisms governing PMN inflammatory responses, however, remain incompletely understood. Based on our recent study showing a non-metabolic role of pyruvate kinase type M2 (PKM2) in controlling PMN degranulation of secondary and tertiary granules and consequent chemotaxis, here we tested a hypothesis that *Pkm2*-deficient mice may resist ALI due to impaired PMN inflammatory responses. We found that PMN aerobic glycolysis controlled the degranulation of secondary and tertiary granules induced by fMLP and PMA. Compared to WT PMNs, *Pkm2*-deficient (*Pkm2^-/-^*) PMNs displayed significantly less capacity for fMLP- or PMA-induced degranulation of secondary and tertiary granules, ROS production, and transfilter migration. In line with this, myeloid-specific *Pkm2^-/-^* mice exhibited impaired zymosan-induced PMN infiltration in the peritoneal cavity. Employing an LPS-induced ALI mouse model, LPS-treated *Pkm2^-/-^* mice displayed significantly less infiltration of inflammatory PMNs in the alveolar space and a strong resistance to LPS-induced ALI. Our results thus reveal that PKM2 is required for PMN inflammatory responses and deletion of PKM2 in PMN leads to an impaired PMN function but protection against LPS-induced ALI.

## 1. Introduction

Polymorphonuclear neutrophils (PMNs) are a double-edged sword, playing essential roles in both the defense against foreign invaders and causation of inflammation-related tissue damage. As a major component of inflammatory and immunological reactions, PMNs play an essential role in acute lung injury (ALI) [1,2]. A significant number of inflammatory PMNs are detected in the lower respiratory tract after lung injury, trauma, or infection. Increased recruitment and over-reactivity of neutrophils in the lungs induce activated alveolar macrophages, influencing ALI severity and development [3]. Activated neutrophils in the lungs produce numerous cytotoxic substances, such as granular enzymes, reactive oxygen species (ROS), various proinflammatory cytokines, and neutrophil extracellular traps (NETs), which trap pathogens in the extracellular space through NETosis [4]. It has been suggested that every aspect of PMN inflammatory responses, including adhesion, diapedesis, reactive oxygen metabolite production, and release of lytic enzymes, is dependent on the orderly mobilization and secretion of various granules and secretory vesicles [5]. Some adhesion molecules in these granules can translocate to the cell surface following PMN activation. For instance, inflammatory stimuli trigger the mobilization of CD11b/CD18-containing secretory vesicles and secondary granules to the cell surface [6,7]. Translocation of CD11b/CD18 to the surface of PMNs mediates firm adhesion of PMN to the epithelial or endothelial monolayers and initiates PMN migration across these cell monolayers through interactions with the intercellular adhesion molecule 1 (ICAM-1) [8].

In our recent study, we reported a non-metabolic role of PKM2 in modulating the degranulation of PMN secondary and tertiary granules through phosphorylation of SNAP-23, which is specifically located at PMN secondary and tertiary granules [9]. Given that degranulation of PMN secondary and tertiary granules is key for PMN migration, we postulate that impaired degranulation of secondary and tertiary granules leads to a defect in PMN infiltration and thus a protective effect on ALI. To validate this hypothesis, in the present study, we generated PMN-specific *Pkm2*-knockout mice, and examined the PMN infiltration and disease index in both WT and *Pkm2*-knockout mice using different inflammatory mouse models. Our results confirm that PKM2 is required for PMN inflammatory responses, and deletion of PKM2 leads to an impaired PMN function but a protective effect against LPS-induced ALI.

## 2. Materials and Methods

### 2.1. Cells

The use and handling of human blood samples in this study was approved by the Institutional Review Boards of the China Pharmaceutical University, and written informed consent was obtained from each healthy volunteer. Human PMNs were isolated from the whole blood of volunteers as previously described [10]. The activation status and chemotactic efficacy of the isolated PMNs were examined using a rapid assay [7]. To obtain mouse bone marrow cells, femur bone cavities were opened by cutting off both ends of the bone, and flushed out using Hank’s buffer devoid of Ca^2+^ and Mg^2+^ followed by the lysis of red blood cells. In certain experiments, the bone marrow leukocytes were further separated using Percoll to obtain a population of mature PMNs.

### 2.2. Mice

Myeloid-specific *Pkm2^-/-^* mice were generated as previously described [9]. Briefly, B6.129S-Pkmtm1.1- Mgvh/J transgenic mice were crossed with B6/JNju-Lyz2^em1Cin(iCre)^/Nju transgenic mice from the Model Animal Research Center of Nanjing University (Nanjing, China). Mouse genotypes were determined by PCR with primers for Lyz2-Cre (Lyz2-iCre-tF2: 5′-AGTGCTGAAGTCCATAGATCGG-3′, Lyz2-iCre-tR2: 5′-GTCACTCATGCTC CCCTGT-3′), and *Pkm2*-loxP (forward primer 5′-AGGTAGGAGGCGGCGTG-3′, reverse primer: 5′-CCACTCACTCTT GGCATCC-3′). All mice were backcrossed to C57BL/6J for at least 10 generations prior to this study. We performed all animal experiments according to the guidelines of the Institutional Animal Care and Use Committee at the China Pharmaceutical University. Male *Pkm2^-/-^* mice and their WT littermates aged 6–8 weeks with a comparable body weight were used for the experiments.

### 2.3. PMN Degranulation Assays

To test the effect of the inhibitors on resting PMNs, cells (5 × 10^6^) were incubated in 400 µL of HBSS with different inhibitors or the same dilution of DMSO at 25 °C for 30 and 60 min. Cell-free supernatants were collected and assayed lactoferrin (LTF) and matrix metallopeptidase 9 (MMP9), correlating to the release of secondary and tertiary granules, respectively [10]. Lactoferrin (Abcam, ab108882, Cambridge, UK) and MMP9 (Abcam, ab100610) were assayed by ELISA. To assay PMN degranulation after chemoattractant stimulation, after treatment with aerobic glycolysis inhibitors and activators for 30 min, including 2-deoxyglucose (2 mM, Sigma, D8375, Kawasaki, Japan), shikonin (1 μM, Sigma, S7576), oleanic acid (OA) (10 μM, Sigma, O5504), serine (5 mM, Sigma, S4500), and FBP (500 µM, Sigma, F6803), or no treatment, PMNs were stimulated with 1 µM fMLP (Sigma, F3506) in HBSS for 20 min (37 °C), followed by assaying for the release of granular markers. Positive degranulation controls were performed by stimulating the same amount of PMNs with fMLP plus 10 µM cytochalasin B (CB) (Sigma, C6762) [11]. Mouse PMNs from bone marrow were obtained with Percoll to assay the levels of LTF and MMP9 using ELISA (USCN, SEA780Mu, and SEA533Mu, Wuhan, China) after fMLP and PMA (Sigma, P1585) stimulation.

### 2.4. Cell-Based Assay for Glucose Uptake and Lactate Production

The levels of 2-NBDG glucose taken up by freshly stimulated PMNs were measured with a Glucose Uptake Cell-Based Assay Kit (Cayman Chemical, 600470-1, Ann Arbor, MI, USA). Cells were seeded in 96-well plates at a density of 4 × 10^4^ cells per well. After a different stimulus was used for the indicated time, glucose uptake assays were performed according to the manufacturer’s protocol, and all experimental groups were compared with the control and presented as a percentage. The levels of lactate production were examined with a Lactate Assay Kit (Biovision, K607, Milpitas, CA, USA) in accordance with the manufacturer’s instructions. The isolated neutrophils (1 × 10^6^) were seeded on a 24-well plate in serum-free RPMI-1640 culture media and then incubated with fMLP and PMA for 30 and 60 min, respectively. Lactate assays were performed with culture media collected from each sample and the optical density was measured at 570 nm using a VARIOSKAN FLASH (Thermo, Waltham, MA, USA).

### 2.5. Western Blot

Cells and mouse tissue from *Pkm2^-/-^* mice and their WT littermates were homogenized in RIPA lysis buffer (Beyotime, P0013C, Nantong, China) supplemented with proteinase and phosphatase inhibitor (Cell Signaling Technology, 5872, Danvers, MA, USA). Proteins were boiled for 10 min at 95 °C with loading buffer, subjected to 10% SDS–PAGE, and then transferred to PVDF membranes and incubated with anti-PKM2 (diluted 1:1000 for WB, Cell Signaling Technology, 4053S), anti-PKM2 (Y105) (diluted 1:1000 for WB, Cell Signaling Technology, 3827S), anti-PKM1 (diluted 1:1000 for WB, Proteintech, 15821-1-AP, Chicago, IL, USA), or anti-GAPDH (diluted 1:1000 for WB, ABclonal, AC033, Woburn, MA, USA) overnight at 4 °C followed by horseradish peroxidase (HRP) conjugated secondary antibody at room temperature. Then, visualization with the Western blotting detection system Tanon-5200 was carried out. All uncropped gel data are shown in Appendix A.

### 2.6. Immunofluorescence

Cells were fixed with 4% paraformaldehyde and then permeabilized with 0.01% Triton X-100 for 10 min and subsequently probed with antibodies against PKM2 (diluted 1:100, Cell Signaling Technology) and CD11b (diluted 1:100, R&D, MAB1124, Minneapolis, MN, USA). Then, incubation with fluorescent-tagged secondary antibodies was carried out. Nuclei were stained with DAPI dye. For fluorescence imaging, a Nikon C2 Plus confocal microscope was used.

### 2.7. Flow Cytometry

The intracellular ROS levels of PMN were measured using a Reactive Oxygen Species Assay Kit (Beyotime, S0033S) in accordance with the manufacturer’s instructions. For the PMN surface level of CD11b/CD18 measurements, mouse PMNs isolated from bone marrow, with PMA stimulation, were stained with anti-CD11b (11-0112-41, THERMO) for 30 min on ice in staining buffer, and then measured at 488 nm using BACKMAN Navios. Data were analyzed using FlowJo (v.10.4) (Ashland, OR, USA), and representative plots are presented in Appendix A.

### 2.8. fMLP-Induced PMN Transfilter Migration

Mouse PMNs were isolated from bone marrow using density gradient centrifugation as previously described [12]. Briefly, bone marrow was flushed from femurs and tibias, and red blood cells were lysed with lysis buffer. Single-cell suspension was layered over a separation gradient of Histopaque of 1.119 and 1.007 g/mL, and centrifuged at 2000 rpm for 20 min at 25 °C. The PMN purity was typically >90% as determined by flow cytometry. Mouse PMN chemotaxis was evaluated using collagen-coated (10 μg/cm^2^) 0.33 cm^2^ polycarbonate Transwells^®^ (5 μm pore size; Costar Corp., Washington, DC, USA), as previously described [10,13]. Briefly, the migration of 1 × 10^6^ PMNs added to the upper chamber of the Transwell inserts was induced using a chemotactic gradient of 0.1 µM of fMLP for various time points at 37 °C. Transmigrated PMN in the bottom chamber were quantified by assaying myeloperoxidase (MPO) (Nanjing Jiancheng Bioengineering Institute, A044-1-1, Nanjing, China) as described previously [10].

### 2.9. Zymosan-Induced PMN Peritoneal Infiltration

Saline solution (1 mL) or saline solution containing zymosan (0.5 mg/mL) (Sigma, Z4250) was then instilled intratracheally into male *Pkm2^-/-^* mice and WT littermates. After 3 and 6 h, the mice were euthanized and the peritoneal cavity was rinsed with 10 mL cold PBS containing 1 mM EDTA, and the rinse solution was collected. The total number of cells was counted using a direct counting plate with a microscope.

### 2.10. LPS-Induced ALI

Mice were administered LPS intratracheally to induce ALI [14]. Saline solution (50 μL) or saline solution containing LPS (0.5 mg/kg, Sigma, L2630) was then instilled intratracheally to the mice. At 24 h post-endotoxin injection, the animals were euthanized with ether followed by dislocation of the cervical spine. The broncho alveolar lavage fluid (BALF) was obtained by gently washing the lung cavities with repeated 700 μL saline lavages up to a total volume of 3.5 mL. The lung was subsequently removed to test the MPO activity using biochemical kits (Nanjing Jiancheng Bioengineering Institute, A044-1-1) according to the manufacturer’s protocols. BALF samples were centrifuged at 1500× *g* for 10 min to collect cells and the red blood cells were removed by lysis. Cell populations (mainly PMNs) were identified by morphological examination of the cell smears and stained with the Diff-Quick staining kit (Solarbio, G1540, Beijing, China). The lung wet weight/dry weight (W/D) ratio was measured to determine the degree of lung edema [15]. Briefly, the fresh upper parts of the right lung were washed and weighed to determine the wet weight and then baked in an 80 °C oven for at least 24 h to assess the dry weight.

### 2.11. Histopathological Evaluation

Mouse lungs were fixed in 4% paraformaldehyde for 24 h and embedded in paraffin. Next, lung tissue sections (5 μm) were prepared and stained with hematoxylin and eosin (H&E) for blinded histopathologic measurement. For immunohistochemical (IHC) analysis, tissue sections were incubated with the specific primary antibodies CD11b (diluted 1:100, R&D, MAB1124) or MPO (diluted 1:100; Cell Signaling Tech, 14569S), then incubated with HRP-conjugated secondary antibody (1:300, Beyotime, P0211) for 1 h, and subsequently developed by a diaminobenzidine (DAB) kit (Invitrogen, 34065, Waltham, MA, USA). The tissue sections were observed under a light microscope (OLYMPUS BX51).

### 2.12. ELISA

The levels of IL-1β (LA137704H), IL-6 (LA137702H), and TNF-α (LA137701H) in the serum and BALF were measured using commercial ELISA kits according to the manufacturer’s instructions (Nanjing Lapuda Biotechnology Co., Ltd., Nanjing, China).

### 2.13. Statistical Analysis

Data derived from at least three independent experiments were presented as the mean ± SEM. Statistical comparisons between two groups were performed by Student’s *t-*test. Multiple-group comparisons were determined using two-way ANOVA. *p*-values < 0.05 were considered significant. The Pearson correlation coefficient (R value) was calculated assuming that the linear relationship between variables depending on the data was parametric.

## 3. Results

### 3.1. PMN Degranulation of Secondary and Tertiary Granules Requires Increased Aerobic Glycolysis

To elucidate the mechanisms controlling PMN degranulation, we explored the correlation between degranulation and metabolism in PMNs under inflammatory stimulation. PMNs were isolated from human peripheral blood and treated with fMLP or PMA for various time periods, which can rapidly release secondary (SG) and tertiary (TG) granules [16]. A combination of fMLP and the actin filament disruption reagent cytochalasin B (CB) can induce a robust and complete degranulation of all granules [11,17]; therefore, PMNs were also treated with fMLP plus CB to serve as positive controls for degranulation. We found that both fMLP (1 μM) and PMA (0.1 μM) induced the release of secondary (SG) (Figure 1A, upper) and tertiary (TG) granules (Figure 1A, lower) in a time-dependent manner. Previous studies suggested that inflammatory stimuli could alter the leukocyte’s metabolic status and increase aerobic glycolysis [18,19]. We next assessed the glucose uptake and lactate production in PMNs before and after fMLP or PMA stimulation. As shown in Figure 1B, fMLP or PMA stimulation also increased glucose uptake and lactate production in PMNs in a time-dependent manner. These results confirm that fMLP or PMA treatment can enhance aerobic glycolysis in PMNs, which is consistent with previous findings that showed chemoattractant stimulates human PMN aerobic glucose metabolism. Interestingly, degranulation of SG and TG was positively correlated with aerobic glycolysis in PMNs (Figure 1C). When aerobic glycolysis in PMNs was blocked by 2-deoxyglucose (2-DG), shikonin [20], oleanic acid (OA) [21], fructose 1,6-bisphosphate (FBP) [22], or serine (Ser) [23], the degranulation of SG and TG by fMLP was also strongly reduced (Figure 1D). Moreover, the phosphorylation of PKM2 significantly increased after fMLP or PMA stimulation, instead of PKM2 (Figure 1E). Given that shikonin, FBP, and Ser have been shown to be inhibitors of PKM2 activity as a protein kinase [24], our results suggest that PKM2, especially phosphorylated PKM2, switched aerobic glycolysis may play a critical role in promoting degranulation of SG and TG in fMLP- or PMA-stimulated PMNs, which is in agreement with our previous studies showing that PKM2 regulates degranulation of neutrophil SG and TG [9].

### 3.2. PKM2-Deficient PMNs Have Impaired Transfilter Migration and Infiltration in Zymosan-Induced Mouse Peritonitis

To further analyze the role of PKM2 in controlling PMN degranulation of SG and TG in vivo, we crossed B6.129S-Pkmtm1.1Mgvh/J transgenic mice with B6/JNju-Lyz2^em1Cin(iCre)^/Nju transgenic mice to generate transgenic mice lacking myeloid-specific *Pkm2* (*Pkm2^-/-^*) (Figure 2A). Western blotting and immunofluorescence staining clearly indicated that *Pkm2* was deleted in mature PMNs but not non-myeloid cells in *Pkm2^-/-^* mice (Figure 2B–D and Appendix A). Furthermore, we found a compensatory increase in PKM1 in myeloid tissues from *Pkm2^-/-^* mice (Appendix A).

To investigate whether PMN function was affected in *Pkm2^-/-^* mice, consistent with previous reports, we found that both fMLP and PMA induced the release of secondary (SG) (Figure 3A, upper) and tertiary (TG) granules (Figure 3A, lower) in mouse PMNs, and *Pkm2^-/-^* PMNs significantly reduced degranulation under stimulation. Moreover, *Pkm2^-/-^* PMNs significantly reduced lactate production during stimulation, as shown in Appendix A. Furthermore, fMLP and PMA resulted in the production of high amounts of ROS from PMN, as shown in Figure 3B. *Pkm2^-/-^* PMNs had significantly reduced ROS levels under stimulation. This is in agreement with previous studies that found cell surface incorporation of CD11b/CD18 as a hallmark of degranulation of SG and TG [6,7] and decreased surface levels of CD11b/CD18 in fMLP-stimulated *Pkm2^-/-^* PMNs [9]. As shown in Figure 3C, PMA-induced surface incorporation of CD11b/CD18 in *Pkm2^-/-^
*PMNs was also defected. Given that cell surface CD11b/CD18 is required for PMN chemotaxis, we next measured the chemotaxis capacity of WT and *Pkm2^-/-^* PMNs using the transfilter migration assay, and the result showed that fMLP-induced transfilter migration of *Pkm2^-/-^* PMNs was significantly slowed down compared to WT PMNs (Figure 3D).

We further studied the influx of WT and *Pkm2^-/-^* PMNs into the peritoneum using a zymosan-induced mouse acute peritonitis model [25,26]. In this experiment, WT and *Pkm2^-/-^* mice were treated with zymosan particles as previously described [9] and peritoneal PMNs were collected and quantified at 3 and 6 h post-zymosan treatment. In line with the finding that *Pkm2^-/-^* PMNs have an impaired chemotactic ability, significantly less PMN infiltration into the peritoneum of *Pkm2^-/-^* mice was detected compared to that in WT mice at 3 and 6 h post-zymosan treatment (Figure 3E).

### 3.3. Myeloid-Specific PKM2 Deficiency Protects Mice from LPS-Induced ALI

Mice treated with LPS, which shows the link between inflammatory responses and hemorrhagic tissue injury, have been widely used as an animal model of ALI [14]. To characterize the role of myeloid-specific *Pkm2*^-/-^ in regulating PMN-mediated ALI, we employed an LPS-induced ALI mouse model and characterized the mouse lung tissue damage by assaying various parameters [27]. As shown in Figure 4A, H&E staining clearly indicated that *Pkm2^-/-^* mice had less lung tissue damage induced by LPS compared to WT mice that underwent the same treatment. In line with this, immunohistochemical staining of CD11b and MPO showed that *Pkm2^-/-^* mice had significantly less infiltration of CD11b- and MPO-positive PMNs in the alveolar space of LPS-treated *Pkm2^-/-^* mice compared with that of WT mice that underwent the same treatment (Figure 4B), and the MPO activity was also significantly lower in the lung tissues from *Pkm2^-/-^* mice (Figure 4C). Supporting this possibility, we found that fewer PMNs were recruited to the BALF of *Pkm2^-/-^* mice compared to that in WT mice at 24 h post-LPS treatment (Figure 4D). Moreover, as shown in Figure 4E, *Pkm2^-/-^* mice exhibited a markedly lower W/D ratio than WT mice following LPS treatment at 24 h after endotoxin injection. Furthermore, our results suggested that LPS-induced secretion of IL-1β, IL-6, and TNF-α levels in serum and BALF were effectively reduced in *Pkm2^-/-^* mice (Figure 4F and Appendix A). In conclusion, the results suggest that myeloid-specific *Pkm2* deficiency attenuates the acute lung edema induced by LPS treatment.

## 4. Discussion

PMN infiltration into the extravascular compartments of the lungs is a hallmark of ALI. When PMNs migrate, they are activated to engulf invading pathogens, releasing proteases, granules, oxidants, and neutrophil extracellular traps, which all play a role in eliminating pathogens. In ALI, PMNs and their cytotoxic substances can cause tissue injury, including an increase in lung endothelial permeability and epithelial apoptosis, which leads to an influx of protein-rich alveolar edema and arterial hypoxemia [28]. In fact, the severity of ALI is associated with increased lung PMNs. Therefore, it is important to understand the mechanisms by which the PMN effector functions in ALI.

Previous studies demonstrated that PKM2 has two different states: the tetramer state with pyruvate kinase activity and the dimer state with protein kinase activity, which is primarily due to the phosphorylation of Y105 [24]. The non-metabolic function of phosphorylated PKM2 has been extensively reported [29,30]. In immune cells, PKM2 has been reported to bridge metabolic and inflammatory dysfunction under various pathophysiologic conditions, such as coronary artery disease [31] and autoimmune conditions [32]. Xie et al. found that PKM2-dependent glycolysis promoted NLRP3 and AIM2 inflammasome activation in macrophages [33]. Our recent study also showed that phosphorylated PKM2 in PMN can serve as a protein kinase to phosphorylate SNAP-23, which is mainly located at secondary and tertiary granules, resulting in degranulation of PMN secondary and tertiary granules [9]. In line with our finding, Dhanesha et al. reported that PKM2 promoted PMN activation and cerebral thromboinflammation [34].

Based on these findings, here we examined the relationship between aerobic glycolysis and PMN degranulation, and the result showed that aerobic glycolysis was associated with PMN degranulation (Figure 1A–C). Tan et al. found that aerobic glycolysis inhibited PMN influx to the infectious site and led to an impaired chemotaxis induced by lipopolysaccharides (LPSs) [35]. We also detected significantly increased p-PKM2 in PMN after fMLP or PMA treatment though the total PKM2 level did not change (Figure 1E). All these results suggest a non-metabolic function of PKM2 in modulating PMN degranulation. In line with this, Zhang et al. found that shikonin, which targets PKM2, could prevent LPS-induced acute lung injury [36].

Given that PMN inflammatory responses, particularly chemotaxis, are dependent on orderly mobilization and degranulation of various granules [37], in this study, we examined PMN infiltration in different inflammatory models with or without a PMN-specific *Pkm2* deficiency. First, we crossed PMN-specific depletion of *Pkm2* mice through the Cre-loxp system, and the result demonstrated myeloid PMN knocked out PKM2, rather than other cells (Figure 2 and Appendix A). Second, in the PMN-specific *Pkm2*^-/-^ mice, we found a compensatory increase in PKM1, suggesting that the metabolic role of pyruvate kinase might not be reduced (Appendix A). However, both the in vitro and in vivo experiments confirmed the non-metabolic role of PKM2 in modulating PMN chemotaxis under inflammatory conditions. *Pkm2^-/-^* PMNs significantly reduced SG and TG degranulation in mouse PMNs treated with fMLP or PMA (Figure 3A). Given that chemotaxis receptors and β_2_ integrins, particularly CD11b/CD18 [38], that are critical for PMN chemotaxis are mainly expressed in SG and TG, orderly degranulation of SG and TG is a key step for the incorporation of these molecules into the cell surface to enable PMN chemotaxis. Our flow cytometry data showed that the cell surface CD11b/CD18 level was markedly reduced in *Pkm2*^-/-^ PMNs (Figure 3C), confirming an impaired chemotactic capacity of *Pkm2*^-/-^ PMNs. As ROS is also a product in SG and TG, we also tested the release of ROS by PMNs and found that *Pkm2* deficiency resulted in significantly less ROS production in PMNs after fMLP and PMA stimulation (Figure 3B). This observation is in agreement with the previous finding that PKM2 inhibition blocked ROS production in macrophages [39]. As PMN degranulation is accompanied by the release of MPO, ROS, and high expression of CD11b [38], we further explored whether PKM2 affects PMN inflammatory responses in vivo. As shown in Figure 3, the *Pkm2* deficiency in PMNs resulted in an impaired PMN chemotaxis induced by fMLP or PMA. Mice with a PMN-specific *Pkm2* deficiency also displayed significantly less tissue infiltration of PMNs under inflammation conditions but strong protection against LPS-induced ALI (Figure 4).

These results suggest that knockout of PKM2 in neutrophils can inhibit the hyperactivation of neutrophils in vitro and in vivo. In the absence of proinflammatory stimulation or inflammatory conditions, PMN-specific *Pkm2*-KO mice displayed no significant differences in the number and chemotaxis of neutrophils compared to WT mice, suggesting that the non-metabolic function of PKM2 in PMNs is mainly associated with inflammatory stimulation. This finding is in agreement with our recent report that neutrophil PKM2 was phosphorylated by fMLP and PMA treatment (Figure 1E) [9]. This study identifies PKM2 as a potential therapeutic target for controlling PMN inflammatory responses in acute lung diseases.

## 5. Conclusions

We provide novel evidence that PKM2 is required for PMN inflammatory responses. We showed that PMN aerobic glycolysis controls the degranulation of secondary and tertiary granules. Compared to WT PMNs, *Pkm2*-deficient (*Pkm2^-/-^*) PMNs displayed a significantly less capacity for fMLP- or PMA-induced degranulation of secondary and tertiary granules, ROS production, and chemotaxis. Lastly, PMN-specific *Pkm2^-/-^
*mice exhibited impaired PMN infiltration in inflamed tissues but strong resistance to LPS-induced ALI.

## Figures and Tables

**Figure 1 biomedicines-10-01193-f001:**
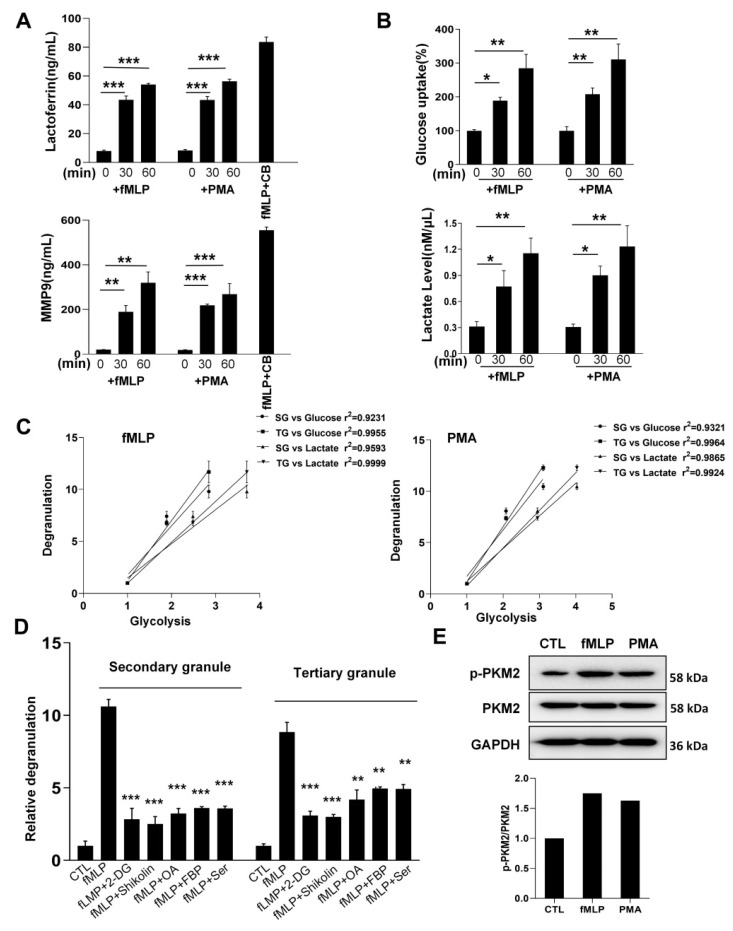
PMN aerobic glycolysis controls the degranulation of secondary and tertiary granules. Human PMNs were isolated from peripheral blood and treated with fMLP (10^−6^ M) and PMA (10^−7^ M) for various durations. (**A**) Degranulation of secondary and tertiary granules represented by the release of lactoferrin and MMP9, respectively. (**B**) Glucose uptake and lactate production during PMN degranulation. (**C**) Correlation between fMLP- and PMA-induced aerobic glycolysis and degranulation of secondary and tertiary granules. (**D**) Inhibition of fMLP- or PMA-induced degranulation of secondary and tertiary granules by aerobic glycolysis inhibitors. (**E**) Enhanced phosphorylation of PKM2 in human PMN by fMLP or PMA stimulation. Data are represented as mean ± SEM. NS, No significance. *, *p* < 0.05. **, *p* < 0.01. ***, *p* < 0.001.

**Figure 2 biomedicines-10-01193-f002:**
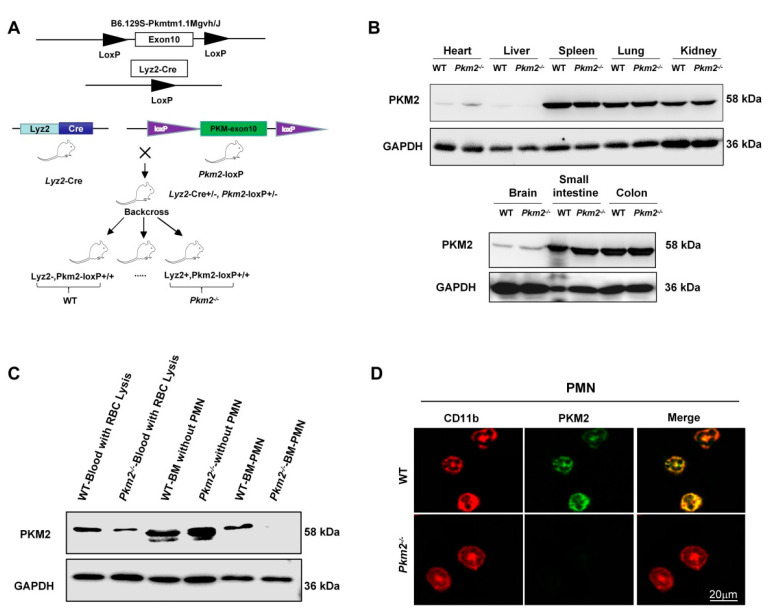
Identification of PMNs from myeloid-specific *Pkm2^-/-^* mice. (**A**) Experimental design of the generation of PMN-specific *Pkm2* knockout mice. (**B**) Validation of PMN-specific *Pkm2* knockout in the non-myeloid tissue of mice using Western blotting. (**C**) Validation of PMN-specific *Pkm2* knockout in the myeloid tissue of mice using Western blotting. (**D**) Validation of PMN-specific *Pkm2* knockout in mice using immunofluorescence labeling.

**Figure 3 biomedicines-10-01193-f003:**
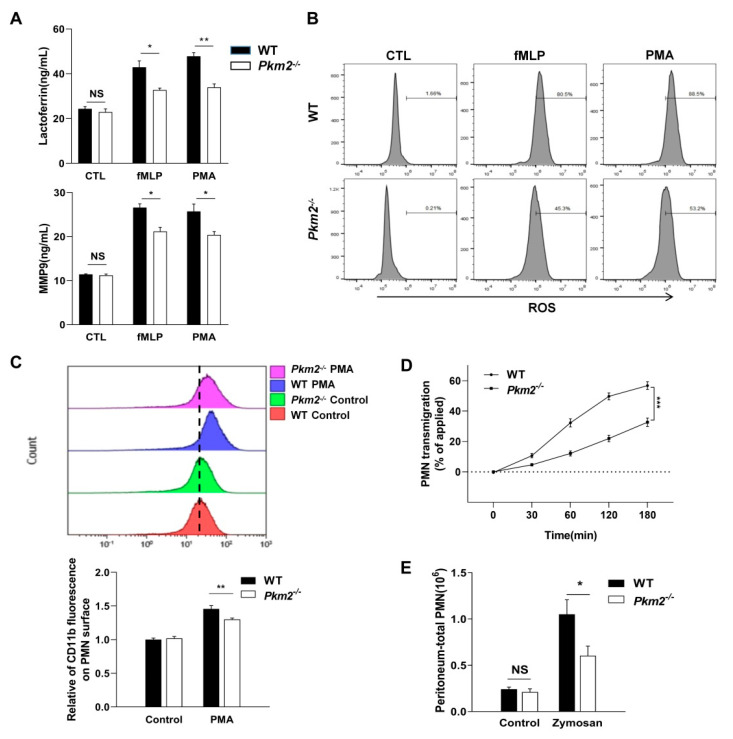
Myeloid-specific *Pkm2-*deficient mice showed less PMN infiltration. (**A**) Degranulation of secondary and tertiary granules represented by the release of lactoferrin and MMP9 in PMN isolated from WT and *Pkm2^-/-^*, respectively. (**B**) ROS production in neutrophils following fMLP and PMA induction. (**C**) Impaired degranulation of secondary/tertiary granules in PMN isolated from *Pkm2^-/-^* mice with PMA induction. Degranulation of secondary/tertiary granules was detected by measuring the surface CD11b level. (**D**) Impaired fMLP-induced trans-filter migration in PMN isolated from *Pkm2^-/-^* mice as measured by the MPO optical density (OD). (**E**) Impaired zymosan-induced PMN peritoneal infiltration in *Pkm2^-/-^
*mice. Data are represented as mean ± SEM. NS, No significance. *, *p* < 0.05. **, *p* < 0.01. ***, *p* < 0.001.

**Figure 4 biomedicines-10-01193-f004:**
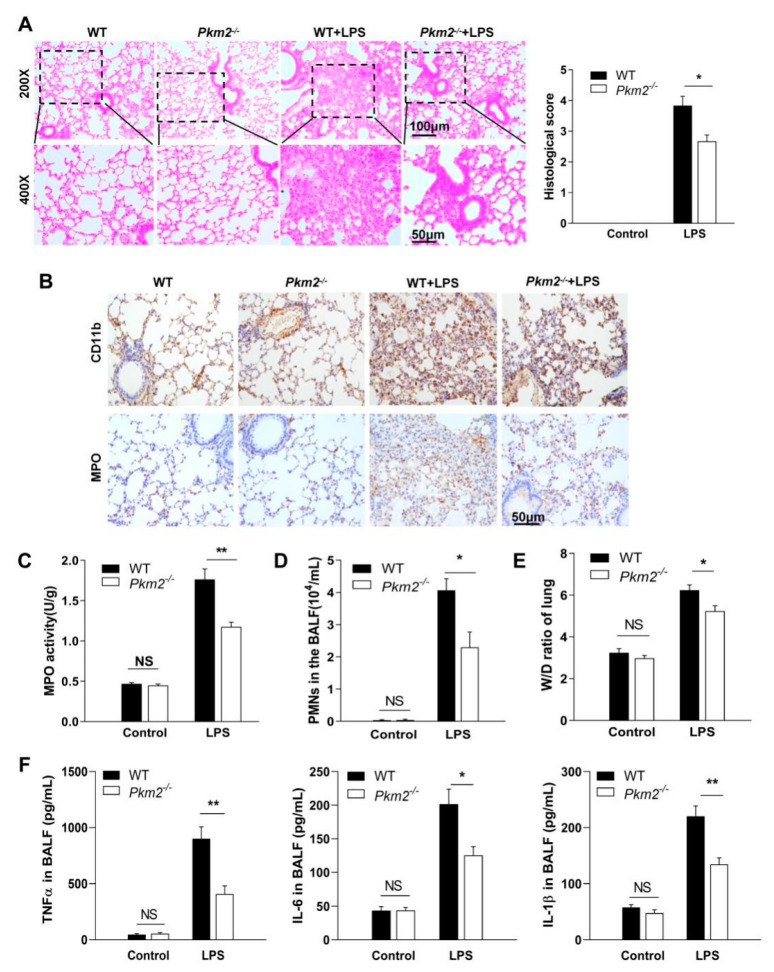
Myeloid-specific *Pkm2* deficiency protects mice against LPS-induced ALI. (**A**) Left, representative H&E staining of lung tissue sections in mice with or without exposure to intratracheal LPS. Right, the histology score of H&E staining. Lung injury was scored by examining the following parameters in a double-blind manner: pulmonary edema, inflammatory infiltration, hemorrhage, atelectasis, and hyaline membrane formation. *n* = 5–6. For each slide, 3–5 views were analyzed. (**B**) IHC staining of CD11b and MPO in mouse lungs with or without LPS treatment. (**C**) MPO level in mouse lungs with or without LPS treatment. (**D**) PMNs recovered in the BALF of mice after 24 h exposure to intratracheal LPS. (**E**) Mouse lung W/D ratio at 0 or 24 h post-LPS treatment. (**F**) The IL-1β, IL-6, and TNF-α contents in BALF were measured using ELISA analysis. Data are represented as mean ± SEM. * *p* < 0.05, ** *p* < 0.01.

## Data Availability

All data and materials support the reported claims and comply with standards of data transparency. Data will be made available on reasonable request.

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
