# Peer review of "Myeloid-Specific Pyruvate-Kinase-Type-M2-Deficient Mice Are Resistant to Acute Lung Injury"

_biomedicines, 2022, doi:10.3390/biomedicines10051193_

Round 1

Reviewer 1 Report

In this work, the authors propose that aerobic glycolysis is essential to promote PMN degranulation under inflammation and, therefore, to induce pulmonary infiltration capacity. Pkm2-deficient (Pkm2-/-) PMNs displayed significantly less capacity of fMLP or PMA-induced degranulation of secondary and tertiary granules, ROS production, as well as transfilter migration. Moreover, LPS-treated Pkm2-/- mice displayed less infiltration of PMN and strong protection against ALI compared with WT mice. This is an interesting work; however, there are major issues that need to be addressed before its publication.

  • The authors show in figure 1 that degranulation correlates with aerobic glycolysis. They use a series of glycolysis blockers and see a decrease in degranulation. If aerobic glycolysis controls degranulation, mice deficient in PKM2 should show impaired degranulation. Please, measure MMP9, lactoferrin, and lactate production in PMNs from PKM2 -/- mice. Pharmacological inhibition of PKM2 through shikonin on WT mice would also help to confirm the results.
  • Have the authors tested PKM1 in the PMN of Pkm2 knockout mice? Is important to rule out a compensatory expression of PKM1.
  • The discussion section needs to be improved for comprehensively discussing results and related references.
  • There are many abbreviations throughout the text that are not explained. When an abbreviation is used for the first time, please describe it. For example, MMP9, CB, BALF, etc. They should also go on the figures´ legends.
  • Human PMNs: please, indicate if they were from healthy volunteers. Moreover, the ethical approval number for both human and mouse studies should be included.
  • Immunofluorescence: Please, describe the microscope used and supplement the data with proper negative controls of the fluorescent stainings.
  • Please describe the cytometer used and add representative plots in the supplementary material.
  • Ether should not be used under any circumstances for the euthanasia of rodents. It is dangerous, slow-acting, and an irritant.
  • Histopathological evaluation. How long were the lungs fixed in paraformaldehyde? And, please, although obvious, add that the tissue sections were observed under a light microscope.
  • Please provide catalog and lot numbers
  • Statistical analysis. Indicate the correlation coefficients used, Pearson or Spearman, depending on whether the data were parametric or not.
  • Figure 1B. When the results are shown in percentages, it is necessary to indicate to which quantity and units 100% corresponds.
  • Figure 1E. Please indicate in the figure legend that the numbers appearing below the plots are the results of the densitometric analysis of the bands. A graph with pPKM2/PKM2 ratio would be more clarifying.
  • Figure 1D. Replace “specific” with “secondary”.
  • Have you tested cytokines in lung tissues?
  • Have you tested IgM levels on BAF to analyze pulmonary vascular leakage?

Author Response

Point 1: The authors show in figure 1 that degranulation correlates with aerobic glycolysis. They use a series of glycolysis blockers and see a decrease in degranulation. If aerobic glycolysis controls degranulation, mice deficient in PKM2 should show impaired degranulation. Please, measure MMP9, lactoferrin, and lactate production in PMNs from PKM2 -/- mice. Pharmacological inhibition of PKM2 through shikonin on WT mice would also help to confirm the results.

Response 1:We greatly appreciate the reviewer’s constructive comment. Accordingly, we have examined MMP9, Lactoferrin, and lactate production in mouse neutrophils (new Figure 3A and Supplementary Figure S4), and the results showed that PKM2 knock-out significantly inhibited the production of lactate and degranulation in PMN. As PKM2 inhibitor shikonin has been shown to prevent LPS-induced acute lung injury (PMID: 29243243), we agree with reviewer that inhibition of PKM2 through shikonin further supports our results. We have incorporated these results as well as the cited references into the revision (the changed parts were highlighted in bright yellow).

Point 2: Have the authors tested PKM1 in the PMN of Pkm2 knockout mice? Is important to rule out a compensatory expression of PKM1.

Response 2: We found that PKM1 was increased in PKM2-knockout mice, which is in agreement with previous findings. This result suggests that increased PKM1 may only compensate the pyruvate kinase activity due to loss of PKM2. However, the effect of PKM2 on neutrophil inflammatory responses (degranulation and chemotaxis) is not due to pyruvate kinase activity but non-metabolic function. In specific, phosphorylated PKM2 serves as a protein kinase to phosphorylate SNAP23, which promotes degranulation of neutrophil secondary granules. Supporting this conclusion is that, when neutrophils were treated with fMLP or PMA, we only found that the increased level of phosphorylated PKM2 but not whole PKM2. The phosphorylation of PKM2, which displays in dimer form with protein kinase (non-metabolic) function in neutrophil has been reported by other investigators (PMID: 34529778, PMID: 34155337). We have cited these papers in the revision (the changed parts were highlighted in bright yellow).

Point 3: The discussion section needs to be improved for comprehensively discussing results and related references.

Response 3: We have revised the discussion section by analyzing the results and citing more related references (Page 320-363, the changed parts were highlighted in bright yellow).

Point 4: There are many abbreviations throughout the text that are not explained. When an abbreviation is used for the first time, please describe it. For example, MMP9, CB, BALF, etc. They should also go on the figures´ legends.

Response 4: We have described the abbreviations at the first time mentioned (the changed parts were highlighted in bright yellow).

Point 5: Human PMNs: please, indicate if they were from healthy volunteers. Moreover, the ethical approval number for both human and mouse studies should be included.

Response 5: Indeed, human PMNs were from healthy donor and we have included ethical approval number for both human and mouse studies (in line 402 and 403, the changed parts were highlighted in bright yellow).

Point 6: Immunofluorescence: Please, describe the microscope used and supplement the data with proper negative controls of the fluorescent stainings.

Response 6: We appreciate reviewer’s comment. Accordingly we have described the microscope used and supplement the data with proper negative controls of the fluorescent stainings (new Supplementary Figure 1) (the changed parts were highlighted in bright yellow).

Point 7: Please describe the cytometer used and add representative plots in the supplementary material.

Response 7: We have described the cytometer used in Materials and Methods in line 135-136 and add representative plots in new Supplementary Figure S3. We also revised the manuscript accordingly (the changed parts were highlighted in bright yellow).

Point 8: Ether should not be used under any circumstances for the euthanasia of rodents. It is dangerous, slow-acting, and an irritant.

Response 8: We apologize for the missing information, and the specific way mice being euthanized is through anesthetization first with ether followed by dislocating the cervical spine. We have revised this in the manuscript (the changed parts were highlighted in bright yellow).

Point 9: Histopathological evaluation. How long were the lungs fixed in paraformaldehyde? And, please, although obvious, add that the tissue sections were observed under a light microscope.

Response 9: The lungs tissue was fixed in paraformaldehyde for 24 h. We have revised this in line 172.

Point 10: Please provide catalog and lot numbers

Response 10: We have inserted the catalog in the manuscript (the changed parts were highlighted in bright yellow), and also provided the information of other antibodies and reagents in the new Supplementary Table 1.

Point 11: Statistical analysis. Indicate the correlation coefficients used, Pearson or Spearman, depending on whether the data were parametric or not.

Response 11: We have indicated the correlation coefficients using Pearson in the section of statistical analysis (the changed parts were highlighted in bright yellow).

Point 12: Figure 1B. When the results are shown in percentages, it is necessary to indicate to which quantity and units 100% corresponds.

Response 12: We have indicated to which quantity and units 100% corresponds in the revision.

Point 13: Figure 1E. Please indicate in the figure legend that the numbers appearing below the plots are the results of the densitometric analysis of the bands. A graph with pPKM2/PKM2 ratio would be more clarifying.

Response 13: We have added a graph with pPKM2/PKM2 ratio in the revision.

Point 14: Figure 1D. Replace “specific” with “secondary”.

Response 14: We have replaced “specific” with “secondary” in Figure 1D as well as in the whole text.

Point 15: Have you tested cytokines in lung tissues?

Response 15: We did not test the level of cytokines in lung tissues. However, we assumed that cytokine levels in lung tissues may have a similar trend of alteration to those in serum and BALF.

Point 16: Have you tested IgM levels on BALF to analyze pulmonary vascular leakage?

Response 16: We did not test IgM levels on BALF. We do agree with reviewer that IgM levels is a better way to analyze pulmonary vascular leakage.

Reviewer 2 Report

The manuscript entitled “Myeloid-specific pyruvate kinase type M2 deficient mice resist to acute lung injury” by Xinlei Sun and colleagues examined the ability of acute lung injury resistance of Pkm2­-deficient mice.  The authors claimed that Pkm-deficient mice exhibit significantly less capacity of fMLP or PMA-induced degranulation of secondary and tertiary granules, ROS production and etc. In addition, Pkm2-/- mice treated with LPS showed less infiltration of inflammatory PMNs in the alveolar space and strong resistance to LPS-induced ALI. The authors concluded that PKM2 is required for PMN inflammatory responses and eliminating PKM2 in PMN leads to impaired PMN function and protection against LPS-induced ALI. However, this manuscript can be improved further:

  • The authors found the correlation between p-PKM2 and aerobic glycolysis in controlling the degranulation of secondary and tertiary granules induced by fMLP and PMA. However, the mechanism underlying p-PKM2 regulating SG and TG degranulation via aerobic glycolysis was not demonstrated in the system studied.
  • I could not find direct evidence that Pkm2 deficiency reduced SG and TG degranulation upon fMLP and PMA treatment.
  • Although authors identified the fewer infiltration of CD11b and MPO in Pkm2-/- mice upon LPS treatment, its relationship with TG and SG degranulation was not illustrated.

Author Response

Point 1: The authors found the correlation between p-PKM2 and aerobic glycolysis in controlling the degranulation of secondary and tertiary granules induced by fMLP and PMA. However, the mechanism underlying p-PKM2 regulating SG and TG degranulation via aerobic glycolysis was not demonstrated in the system studied.

Response1: We greatly appreciate the reviewer’s constructive comment. The potential mechanism by which p-PKM2 regulating SG and TG degranulation via aerobic glycolysis has been addressed in our recent study (PMID: 34155337), in which we found phosphorylated PKM2 can serve as a protein kinase and directly interacts with SNAP-23, leading to phosphorylation of SNAP-23. As SNAP-23 specifically exists at SG and TG in neutrophils, its phosphorylation will promote the fusion of SG and TG with plasma membranes (degranulation). Based on these findings, this paper mainly focused on validating this phenomenon in vivo by using Pkm2-/- mice. To do this, we performed various inflammation simulations in vitro and in vivo, respectively, and the results showed that PMN-specific Pkm2-/- mice have impaired degranulation of SG and TG under inflammatory conditions, and thus display a strong resistance to LPS-induced ALI.

Point 2:I could not find direct evidence that Pkm2 deficiency reduced SG and TG degranulation upon fMLP and PMA treatment.

Response2: Again, we apologize for the missing information. The direct evidence that Pkm2 deficiency reduced SG and TG degranulation in PMN has been reported in our recent study (PMID: 34155337). Nevertheless, to link the PKM2-switched glycolysis with SG and TG degranulation in neutrophils, we have examined MMP9, lactoferrin, and lactate production in mouse neutrophils, and the results showed that PMN-specific PKM2 deficiency significantly inhibited the production of lactate and degranulation in PMN (new Figure 3A and Supplementary Figure S4). We have revised the manuscript accordingly (the changed parts were highlighted in bright yellow).

Point 3: Although authors identified the fewer infiltration of CD11b and MPO in Pkm2-/- mice upon LPS treatment, its relationship with TG and SG degranulation was not illustrated.

Response3: We appreciate the reviewer’s constructive comment. We have discussed this issue in line 352-363 (the changed parts were highlighted in bright yellow). PMN chemotaxis is dependent upon the cell surface chemokine receptor and β2 integrins particularly CD11b/CD18 [reference #39]. Given that chemotaxis receptors and β2 integrins are mainly expressed in SG and TG, therefore it is generally accepted that orderly degranulation of SG and TG is a key step to enable PMN chemotaxis because only through degranulation the chemotaxis receptors and β2 integrins can be incorporated into cell surface. In the present study, we presented evidence that PMN-specific Pkm2 deficiency significantly reduced degranulation of SG and TG in mouse PMNs (Figure 3A), and the production of ROS (another component in SG and TG of neutrophils) after fMLP and PMA stimulation (Figure 3B). Our flow cytometry data also showed that cell surface CD11b/CD18 level was markedly reduced in Pkm2-/- PMNs (Figure 3C), confirming an impaired chemotactic capacity of Pkm2-/- PMNs. MPO was only used as a PMN functional marker to show the level of infiltrated PMNs. 

Round 2

Reviewer 1 Report

I want to congratulate the authors since the paper has significantly improved compared with the first version. This reviewer regrets not having more time to dedicate to this paper but encourages authors to keep working on it. After a quick second revision, the paper still has a few minor mistakes to be corrected

- Authors should indicate the method of euthanasia since in the last version of the article this has been removed.

- There are some misspelled words, please review the entire article.

- Abbreviations should be indicated the first time they appear in the text. “Bronchoalveolar Lavage Fluid” should go on line 158.

- Please, describe the microscope used for the histopathological evaluation.

- Please check the font of the references

Author Response

Point-to-point response

Point 1: Authors should indicate the method of euthanasia since in the last version of the article this has been removed.

Response 1: We have inserted the method of euthanasia in line 159-160 (the changed parts were highlighted in bright green).

Point 2: There are some misspelled words, please review the entire article.

Response 2: We have checked the full text and corrected misspellings with highlighted in bright green, such as Western Blotting, etc.

Point 3: Abbreviations should be indicated the first time they appear in the text. “Broncho-alveolar Lavage Fluid” should go on line 158.

Response 3: We have indicated the abbreviations they first time they appear accordingly. For example, “Broncho-alveolar Lavage Fluid” for BALF was indicated on line 160.

Point 4: Please, describe the microscope used for the histopathological evaluation.

Response 4: We have described the microscope used for the histopathological evaluation in Materials and Methods (line 179-180).

Point 5: Please check the font of the references

Response 5: We have revised the fonts in the references and keep them consistent with the full text.